# Insecticidal Activity of Chitinases from *Xenorhabdus nematophila* HB310 and Its Relationship with the Toxin Complex

**DOI:** 10.3390/toxins14090646

**Published:** 2022-09-18

**Authors:** Jia Liu, Hui Bai, Ping Song, Ziyan Nangong, Zhiping Dong, Zhiyong Li, Qinying Wang

**Affiliations:** 1Institute of Millet Crops, Hebei Academy of Agriculture and Forestry Sciences, National Foxtail Millet Improvement Center, Minor Cereal Crops Laboratory of Hebei Province, Shijiazhuang 050035, China; 2College of Plant Protection, Hebei Agricultural University, Baoding 071000, China

**Keywords:** *Xenorhabdus nematophila* HB310, chitinases, gene knock out, toxin complex, insecticidal activity

## Abstract

*Xenorhabdus nematophila* HB310 secreted the insecticidal protein toxin complex (Tc). The *chi60* and *chi70* chitinase genes are located on the gene cluster encoding Tc toxins. To clarify the insecticidal activity of chitinases and their relationship with Tc toxins, the insecticidal activity of the chitinases was assessed on *Helicoverpa armigera*. Then, the *chi60* and *chi70* genes of *X. nematophila* HB310 were knocked out by the pJQ200SK suicide plasmid knockout system. The insecticidal activity of Tc toxin from the wild-type strain (WT) and mutant strains was carried out. The results demonstrate that Chi60 and Chi70 had an obvious growth inhibition effect against the second instar larvae of *H. armigera* with growth-inhibiting rates of 81.99% and 90.51%, respectively. Chi70 had a synergistic effect with the insecticidal toxicity of Tc toxins, but Chi60 had no synergistic effect with Tc toxins. After feeding Chi60 and Chi70, the peritrophic membrane of *H. armigera* became inelastic, was easily broken and leaked blue dextran. The Δ*chi60*, Δ*chi70* and Δ*chi60*-*chi70* mutant strains were successfully screened. The toxicity of Tc toxins from the WT, Δ*chi60*, Δ*chi70* and Δ*chi60*-*chi70* was 196.11 μg/mL, 757.25 μg/mL, 885.74 μg/mL and 20,049.83 μg/mL, respectively. The insecticidal activity of Tc toxins from Δ*chi60* and Δ*chi70* was 3.861 and 4.517 times lower than that of Tc toxins from the WT, respectively, while the insecticidal activity of Tc toxins from the Δ*chi60*-*chi70* mutant strain almost disappeared. These results indicate that the presence of *chi60* and *chi70* is indispensable for the toxicity of Tc toxins.

## 1. Introduction

Chitin composed of linear β-1,4-N-acetylglucosamine (GlcNAc) residues is a major component of the intestinal peritrophic membrane (annelids and some arthropods) and exoskeleton (arthropods) [1,2,3,4]. Chitinases (EC 3.2.1.14) are a kind of chitin-degrading glycosidase that play an important role in the hydrolysis of glycosidic bonds in chitin to form soluble chitooligosaccharides [5,6,7]. Chitinases are produced by a variety of microorganisms, with diverse structures and functions. The chitinases bind to the chitin in the exoskeleton or peritrophic membrane, which can lead to structural changes and increase the accessibility of the substrate for the pathogens into the haemocoel of susceptible insects [8,9]. In addition, chitinases could also promote the process of toxin binding to specific receptors and be used to improve the insecticidal activity of toxins [8,10]. Therefore, chitinases have been used in agriculture as an effective virulence factor against pests.

The emergence of *Bacillus thuringiensis* (Bt)-resistant insects has made it important to identify other novel biopesticides [11,12]. Toxin complex proteins (Tc) comprise a candidate class of molecules [8,13]. Tc toxins were first identified in *Photorhabdus luminescens* W14 [14,15], which belongs to the Enterobacteriaceae family and lives in a mutualistic symbiosis with entomopathogenic nematodes (EPNs) from the genus *Heterorhabditis* [16,17]. Tc toxins have high molecular weights and multi-subunit protein complexes, which have high insecticidal activity against various pests [18,19]. Tc toxins consist of three separate components: TcA, TcB and TcC [20,21,22,23,24,25,26]. TcA proteins harbor the cytotoxic effects of Tc toxins, while TcB and TcC proteins modulate and enhance the toxicity of TcA proteins [21,27,28].

Tc toxins are found in *P. luminescens* and *Photorhabdus asymbiotica* as well as in other entomopathogenic bacteria, such as *Xenorhabdus nematophila* [18,29], *Serratia entomophila* [30,31] and *Yersinia entomophaga* [15]. In *Y. entomophaga*, two putative chitinases (Chi1 and Chi2) are contained in the 3D structure of Tc toxins [14,15]. Chi1 and Chi2 proteins are vital for this complex formation [14]. Two chitinase genes (chitinase 60 (*chi60*) and chitinase 70 (*chi70*)) were also found in the locus of Tc toxins from *X. nematophila* [14,32]. The relationship between Tc toxins and chitinases is currently unclear.

The *X. nematophila* HB310 is symbiotically associated with a strain of the entomopathogenic nematode *Steinernema carpocapsae* isolated from the soil in Hebei Province, China [33]. In a previous study, the peptides of chitinases from the intracellular proteins of *X. nematophila* HB310 were identified by matrix-assisted laser desorption-time-of-flight mass spectrometry (MALDI-TOFMS). We also found that the recombinant chitinase 60 (Chi60) and chitinase 70 (Chi70) could enhance the insecticidal activity of the Bt HD73 strain and the Bt Cry1Ac toxin. To clarify the relationship between chitinases and Tc toxins, we determined the insecticidal activity of Chi60 and Chi70 and the pathologic effects on the peritrophic membranes of *Helicoverpa armigera* (Lepidoptera: Noctuidae). Then, the *chi60* and *chi70* genes of *X. nematophila* HB310 were knocked out by the pJQ200SK suicide plasmid knockout system. Moreover, the insecticidal activity of Tc toxins from the wild-type strain (WT), *chi60* gene knockout mutants (Δ*chi60*), *chi70* gene knockout mutants (Δ*chi70*), and double gene (*chi60* and *chi70*) knockout mutants (Δ*chi60-chi70*) in *X. nematophila* HB310 were determined against the second instar larvae of *H. armigera*. This research will help reveal the insecticidal mechanism of Tc toxins and lay a foundation for the development and utilization of insecticidal formulations for entomopathogenic nematode symbiotic bacteria. In addition, insecticidal genes can also be cloned from symbiotic bacteria for the development of transgenic insect-resistant crops, thereby delaying the resistance of pests to Bt transgenic insect-resistant crops.

## 2. Results

### 2.1. Sequence Analysis of Chitinases

Chi60 had 535 amino acids with a predicted molecular weight of 59.3 kDa and a PI of 4.51, Chi70 had 648 amino acids with a molecular weight of 72.4 kDa and PI of 4.89. The two amino acid sequences showed low similarity, with a similarity of 24.29% (Figure 1).

The chitinases from *X. nematophila* HB310 and other bacteria were used to construct the phylogenetic tree to assess the evolutionary relationships among the chitinases. Then, a schematic representing the structure of all complete chitinase sequences was constructed from the MEME motif analysis results. As shown in Figure 2, the chitinases were divided into two subclasses. Chitinases in the same subclass usually showed a highly similar motif composition. All chitinases contain motif 1–3, motif 5, motif 8 and motif 9. However, motif 4 was unique to the subclass of Chi70. Compared to Chi60, motif 4 and motif 6 were unique to Chi70. Compared to Chi70, motif 7 and motif 10 were unique to Chi60. In addition, the sequence similarity of the two subgroups was very low, which could perform different biological functions as two different subclasses.

### 2.2. Insecticidal Activity of Chitinases and Synergistic Effect with Tc Toxin

The inhibitory effect of chitinases on the growth of second-instar larvae of *H. armigera* was determined by feeding methods. Chitinases could significantly inhibit the growth of *H. armigera* (Table 1). At the same concentration, the growth inhibition rates of Chi60 and Chi70 were 81.99% and 90.51%, respectively. The growth inhibition rate of Chi70 against *H. armigera* was higher than Chi60. Both Chi60 and Chi70 had a lower lethal effect on *H. armigera*, with a corrected mortality of 13.89% and 4.17%.

In the search for possible synergistic interactions between chitinase and Tc toxins, different combinations were tested against the second-instar larvae of *H. armigera*—the results of which are shown in Table 2. The LC_50_ value of Tc toxins against the second instar larvae of *H. armigera* is 196.11 μg/mL, however, the LC_50_ value of Tc toxins mixed with Chi70 was 146.47 μg/mL. Chi70 had a synergistic effect on Tc toxins, while Chi60 did not exhibit synergistic toxicity to Tc toxins against *H. armigera* with an LC_50_ value of 185.85 μg/mL.

### 2.3. Pathological Effect of Chitinases on the Peritrophic Membrane of H. armigera

In order to clarify the pathological effect of chitinases on the peritrophic membrane of the fifth-instar larvae from *H. armigera*, the effect of chitinases on the damage and permeability of the peritrophic membrane was determined by feeding method. The peritrophic membrane of the fifth-instar larvae of *H. armigera* after phosphate-buffered solution (PBS, pH 7.2) treatment was intact, translucent, elastic and swinging in water without breaking (Figure 3a). After feeding with Chi60 or Chi70, the peritrophic membrane broke into inelastic fragments when it swung in water, and there was obvious tissue fragmentation (Figure 3b,c).

The effect of chitinases on the permeability of the peritrophic membrane from *H. armigera* was determined by feeding methods. The peritrophic membrane of fifth-instar larvae *H. armigera* was intact after PBS (pH 7.2) treatment, and there was no exudation of blue dextran (Figure 4a). The peritrophic membrane of *H. armigera* displayed obvious exudation after feeding Chi60 or Chi70 (Figure 4b,c).

### 2.4. Homologous Recombination Vector Construction

Six DNA fragments were successfully amplified by PCR and cloned using the methods described (Appendix A). The results indicate that the upstream and downstream fragments of the *chi60* gene and *chi70* gene were successfully amplified from the genomic DNA of *X. nematophila* HB310, respectively. The DNA sequencing identified that the size of upstream and downstream fragments from the *chi60* gene were 1069 bp and 1223 bp, and the size of upstream and downstream fragments from the *chi70* gene were 1147 bp and 1081 bp, respectively. The *Kmr* (1300 bp) and *tetA* (1300 bp) were successfully amplified from the plasmids of pYBA-1132 and pTKLP-tet, respectively.

The whole fragments were successfully amplified by fusion PCR (Appendix A). The results indicate that *chi60*-*Kmr* was successfully fused with a size of 3900 bp, and amplified by *Kmr* and the upstream and downstream fragments of the *chi60* gene. The *chi70*-*tetA* was successfully fused with a size of 4175 bp, and amplified by the *tetA* and the upstream and downstream fragments of the *chi70* gene.

Verification of the correctness of the constructed homologous recombination vector by double digestion. The plasmid DNA of pJQ200SK-*chi60*-*Kmr* and pJQ200SK-*chi70*-*tetA* were digested with Xba I and Xho I, generating two fragments (Appendix A). These results indicate that the 3900 bp *chi60*-*Kmr* and 4175 bp *chi70*-*tetA* were successfully amplified by fusion PCR, respectively.

### 2.5. Identification of Single Gene Knockout Mutants

Δ*chi60* and Δ*chi70* were screened by homologous recombination between the recombinant *E. coli* S17-1 λ pir containing the target genes and the WT. As shown in Figure 5, the fragments of Δ*chi60* or Δ*chi70* were smaller than that of the WT, which were amplified by the primer of 60-up-F/60-down-R or 70-up-F/70-down-R, respectively. The results indicate that the *chi60* or *chi70* gene from the WT had been successfully replaced by the resistance gene from the homologous recombination strain, respectively.

### 2.6. Identification of Double Gene Knockout Mutants

Δ*chi60*-*chi70* were screened by homologous recombination between the recombinant *E. coli* S17-1 λ pir containing *chi60*-*Kmr* and Δ*chi70*. Two pairs of primers (60-up-F/60-down-R and 70-up-F/70-down-R) were simultaneously used to detect the mutants. Δ*chi60*-*chi70* was successfully screened (Figure 6).

### 2.7. Western Blot Analysis

To verify the correctness of gene knockout, the Western blot analysis of the Tc toxins from the WT and knockout mutants was performed (Figure 7). Two bands of 78 kDa (Chi70) and 65 kDa (Chi60) were found in Tc toxins from the WT. The band of 65 kDa disappeared when the *chi60* gene was knocked out, and the band of 78 kDa disappeared when the *chi70* gene was knocked out. It was found that both bands of 78 kDa and 65 kDa disappeared when the *chi60* and *chi70* were simultaneously knocked out. Western blot analysis also showed that the *chi60* gene and *chi70* gene were successfully knocked out.

### 2.8. Insecticidal Activity of Tc Toxins

To clarify the relationship of insecticidal activity between chitinases and Tc toxins, the insecticidal activity of Tc toxins from the WT and gene knockout mutants to the second instar larvae of *H. armigera* was tested by feeding method. The LC_50_ of Tc toxins to the second-instar larvae of *H. armigera* was shown in Table 3. The LC_50_ value of Tc toxins from the WT against the second-instar larvae of *H. armigera* was 196.11 μg/mL. The LC_50_ values of Tc toxins from Δ*chi60* and Δ*chi70* were 757.25 μg/mL and 885.74 μg/mL, respectively. The virulence of Tc toxins from Δ*chi60* and Δ*chi70* were significantly lower than that of the WT, and the toxicity of Tc toxins from Δ*chi60*-*chi70* almost disappeared with the LC_50_ of 20,049.83 μg/mL.

## 3. Discussion

The chitinolytic mechanism of bacteria primarily consists of chitinase [17], which specifically degrades chitin and prevents chitin biosynthesis. In this study, chitinases could significantly inhibit the growth of the second-instar larvae of *H. armigera* and damage the peritrophic membrane of *H. armigera*. In previous studies, the chitinase from *Bacillus subtilis* could effectively inhibit the growth of *Spodoptera litura* (Lepidoptera: Noctuidae) [34]. Chitinase purified from *Pseudomonas fluorescens* MP-13 demonstrated 100% mortality against *Helopeltis theivora* (Heteroptera: Miridae) [35]. Among the seven chitinases isolated from *Bacillus firmus*, *Bacillus licheniformis*, *Thermomyces lanuginosus* and *Streptomyces* sp., most of the chitinases can delay the pupation of *Sesamia calamistis* (Lepidoptera: Noctuidae) and *Chilo partellus* (Lepidoptera: Pyralidae) [36]. This indicates that the insecticidal activities of chitinases from different microorganisms have some differences. However, in this study, the inhibition of Chi60 was significantly lower than that of Chi70, which could be related to the inclusion of Chi60 [9]. It is speculated that Chi60 protein becomes a soluble protein after denaturation and renaturation, but the natural conformation could not be completely restored. Therefore, the inhibition of Chi60 in the second-instar larvae of *H. armigera* was lower than that of Chi70.

Chitinases can accelerate the binding process of toxins to the receptors and cause perforation in the gut peritrophic membrane, which increases accessibility to the substrate and makes it easier for pathogens to enter the haemocoel of susceptible insects [8]. In this study, Chi70 had a synergistic effect on the insecticidal activity of Tc toxins from *X. nematophila*. Previously, studies reported that chitinases produced by VLBt27, VLBt38, VLBt109 and VLBt135 strains isolated from more than 80 *B. thuringiensis* strains could enhance the insecticidal activity of insecticides to *H. armigera* and *Brevicoryne brassicae* (Hemiptera: Aphididae) [37]. In addition, chitinases from *B. thuringiensis* could also enhance the insecticidal activity of its crystal protein against *Plutella xylostella* (Lepidoptera: Plutellidae) [38], *Lymantria dispar* (Lepidoptera: Lymantriidae) [39], *Spodoptera exigua* (Lepidoptera: Noctuidae) and *H. armigera* [40]. However, in this study, the synergistic effect of Chi70 on the insecticidal activity of Tc toxins was lower than its synergistic effect on Bt Cry1Ac toxins [8]. This could be due to the action mode of *X. nematophila*, which was carried into the host insect hemocoel depending on the nematode and did not need themselves to destroy the cuticle or peritrophic membrane of the insect midgut. It is speculated that some functions of *X. nematophila* could degenerate during the evolutionary process, while the toxicity of toxins or secondary metabolites could decrease.

The *chi60* and *chi70* genes from *X. nematopbila* are vital to the insecticidal activity of Tc toxins. In a previous study, the insecticidal activity of the Tc toxins disappeared after the knockout of the *chi1* and *chi2* genes in the toxin complex locus of *Y. entomophaga* MH96 [14,41]. In this study, Δ*chi60*, Δ*chi70*, and Δ*chi60*-*chi70* were constructed by homologous recombination using pJQ200SK plasmids containing the *sacB* gene. The insecticidal activities of Tc toxins from the mutant strains were lower than that of the WT when *chi60* or *chi70* were knocked out. However, the insecticidal activity of Tc toxins from Δ*chi60*-*chi70* almost disappeared after *chi60* and *chi70* were simultaneously knocked out. It is speculated that the chitinases (Chi60 and Chi70) from *X. nematophila* and the chitinases (Chi1 and Chi2) from *Y. entomophaga* have similar functions.

The chitinases from *X. nematophila*, *P. luminescens*, *P. asymbiotica* and *Y. entomophaga* MH96 had the same location on the loci of Tc toxins [14]. The chitinases sizes vary widely within 20–90 kDa, and bacterial chitinases had a size range of 20–60 kDa [42]. Chitinases from *X. nematophila* and *Y. entomophaga* all had molecular weights higher than 60 kDa. Based on the data of GH-18 domains in the phylogenetic tree, Chi60 from *X. nematophila* and Chi1 from *Y. entomophaga* were clustered into the same branch, while Chi70 from *X. nematophila* and Chi2 from *Y. entomophaga* were clustered into the same branch [8]. In *Y. entomophaga*, the genetic knockout of *chi1* and *chi2* genes forms no complex even though the remaining genes are still expressed [41]. In this study, the *chi60* and *chi70* genes of *X. nematophila* were successfully knocked out alone and simultaneously, though whether the knockout of the *chi60* and *chi70* genes influenced the formation of the toxin complex must be further verified. The main research at present is the effect of the chitinase gene knockout on the insecticidal activity of Tc toxins. The next step is to further study the effect of chitinase gene knockout on the structure of Tc toxins. In addition, it is also necessary to pay attention to the development of biological pesticides, and effectively develop the Tc toxins into a new biological pesticide and produce it on a large scale. At the same time, further exploration of transgenic insect-resistant plants with Tc toxin genes is needed.

## 4. Conclusions

Chi60 and Chi70 had an obvious growth inhibition effect against the second-instar larvae of *H. armigera*. Chi60 and Chi70 could destroy the peritrophic membrane of the fifth-instar larvae of *H. armigera*. Chi70 had a synergistic effect with the insecticidal toxicity of Tc toxins, but Chi60 had no synergistic effect with Tc toxins.

The Δ*chi60*, Δ*chi70*, and Δ*chi60*-*chi70* were successfully screened using homologous recombination. The insecticidal activity of Tc toxins from WT, Δ*chi60*, Δ*chi70*, and Δ*chi60*-*chi70* were 196.11 μg/mL, 757.25 μg/mL, 885.74 μg/mL and 20,049.83 μg/mL, respectively. The insecticidal activity of Tc toxins from Δ*chi60*-*chi70* almost disappeared.

These results will help reveal the insecticidal mechanism of Tc toxins and lay a foundation for the development and utilization of insecticidal formulations for entomopathogenic nematode symbiotic bacteria.

## 5. Materials and Methods

### 5.1. Insects, Microorganisms and Proteins

*H. armigera* larvae were obtained from the Pest Biocontrol Laboratory, Hebei Agricultural University, China. The larvae were fed with an artificial diet (13% maize meal, 6.5% soybean powder, 5.8% dry yeast, 0.2% sorbic acid, 0.2% methyl-para-hydroxybenzoate, 4.7% vitamin C, 0.2% compound vitamin B, 3.2% sucrose, 1.3% agar and 64.9% sterilized distilled water) and reared at 28 °C and 70% relative humidity (RH) under a 16 h (h) light (L): 8 h dark (D) photoperiod.

*X. nematophila* HB310 was isolated from *Steinernema carpocapsae* HB310, which was screened from the soil in Hebei province of China and stored in the Pest Biocontrol Laboratory, Hebei Agricultural University, China [16,33]. The bacteria were incubated in Luria–Bertani (LB) broth for 48 h at 28 °C on a rotary shaker at 200 revolutions per minute (rpm). 

The pYBA-1132 plasmid, pTKLP-tet plasmid, pJQ200SK plasmid and *Escherichia coli* S17-1 λ pir competent cell were contributed by the researcher Guangyue Li of the Chinese Academy of Agricultural Sciences.

The chitinases (Chi60 and Chi70) and Tc toxins were obtained from the Pest Biocontrol Laboratory, Hebei Agricultural University, China. The concentration of chitinases was diluted to 1000 μg/mL. The concentration of Tc toxins was diluted to 5000 μg/mL.

### 5.2. Sequence Analysis of Chitinase

The ExPASy tools (https://web.expasy.org/compute_pi/, accessed on 1 August 2022) were used to predict the isoelectric points (pI) and molecular weights (MWs) of chitinases. The conserved domains of chitinases were predicted by the hmmsearch tool [43]. The MEME online program for protein sequence (http://meme.nbcr.net/meme/intro.html, accessed on 2 August 2022) was used to identify the conserved motifs of chitinases, which the optimized parameters being any number of repetitions, a maximum number of 10 motifs and an optimum of 6–200 residues. The full-length amino acid sequences of chitinases from different bacteria were aligned using ClustalW with default parameters [44]. After sequence alignments, the phylogenetic tree was constructed by MEGA5.0 software using the neighbor-joining method with the following parameters: Poisson model, pairwise deletion and 1000 bootstrap replications [45]. The protein names and sequences of chitinases that were used in this analysis were listed in Appendix A. All sequences were obtained from NCBI (https://www.ncbi.nlm.nih.gov/, accessed on 2 August 2022).

### 5.3. Assay for Insecticidal Activity of Chitinases and Pathological Effect

#### 5.3.1. Assay for Insecticidal Activity of Chitinase

Chitinase was diluted to 1000 μg/mL and mixed with the artificial diet at a dose of 100 μL protein per gram of artificial diet. The same volume of 10 mM PBS (pH 7.2) (Coolaber, Beijing, China) was used as a control. Approximately 0.1 g artificial diet and one second-instar larva of *H. armigera* were transferred to each well of a 24-well tissue culture plate and then incubated at 28 °C. The corrected mortality and growth inhibition rate were calculated according to the following formulas at 120 h after treatment:

Growth inhibition rate (%) = ((Weight of control − Weight of treatment)/(Weight of control − Weight of initial)) × 100

Corrected mortality (%) = ((Mortality rate of treatment − Mortality rate of control)/(1 − Mortality rate of control)) × 100

The synergistic effect of Chi60 and Chi70 with Tc toxins against the second-instar larvae of *H. armigera* was tested. Tc toxins were diluted to 2000, 1000, 500, 250, 125, 62.5 and 31.25 μg/mL. Chi60 and Chi70 were added to every treatment in the same quantity (1000 μg). The method of treatment and the culture condition were consistent with the above methods. Each treatment was replicated three times (*n* = 72 larvae per concentration). The LC_50_ of Tc toxin and its mixture with Chi60 and Chi70 was calculated at 72 h after treatment.

#### 5.3.2. Pathological Effect of Chitinase on the Peritrophic Membrane

The chitinase (1000 μg) was added to a 10 g artificial diet. The same quantity of 10 mM PBS (pH 7.2) was used as a control. Approximately 1 g artificial diet and one fifth-instar larva of *H. armigera* were transferred to each well of a six-well tissue culture plate and then incubated at 28 °C. The peritrophic membrane was extracted and the artificial diet inside was washed at 48 h after feeding. Then, the washed peritrophic membrane was placed on a concave slide, and the morphology was observed under a dissecting microscope.

The chitinase (1000 μg) and blue dextran 2000 (100 μg) (Solarbio, Beijing, China) were added to a 10 g artificial diet. The method of treatment and the culture conditions were consistent with the above methods. The peritrophic membrane was extracted at 48 h after feeding. Then, the peritrophic membrane was placed on a concave slide, and the morphology was observed under a dissecting microscope.

### 5.4. Knockout of the Chitinase Gene from X. nematophila HB310

#### 5.4.1. Genomic DNA Extraction

The genomic DNA, used as a template for PCR, was extracted from *X. nematophila* HB310 using the Bacterial Genomic DNA Extraction Kit (TIANGEN, Beijing, China). The quality of genomic DNA was detected using a Touch Screen MD2000C Nano-Spectrophotometer (Biofuture, Shanghai, China) prior to 1% agarose gel electrophoresis.

#### 5.4.2. Homologous Recombination Vector Construction

Six pairs of specific primers based on the gene sequence of *chi60* (GenBank access no.: KC701470), *chi70* (GenBank access no.: KC701471), kanamycin resistance gene (*Kmr*) (GenBank access no.: KU221181), and tetracycline resistance gene (*tetA*) (GenBank access no.: KR071151) are listed in Appendix A. Oligo primers 60-up-F/60-up-R and 60-down-F/60-down-R were used to amplify the upstream and downstream fragments of the *chi60* gene. Oligo primers 70-up-F/70-up-R and 70-down-F/70-down-R were used to amplify the upstream and downstream fragments of the *chi70* gene. Oligo primers Kmr-F/Kmr-R and tet-F/tet-R were used to amplify the kanamycin resistance gene (*Kmr*) and tetracycline resistance gene (*tetA*), respectively.

The purified upstream fragment of the *chi60* gene, downstream fragment of the *chi60* gene and the *Kmr* gene were used as a template for the fusion amplification of *chi60-Kmr*. The purified upstream fragment of the *chi70* gene, downstream fragment of the *chi70* gene, and the *tetA* gene were used as a template for the fusion amplification of *chi70-tetA*.

The resulting PCR products were ligated into the pJQ200SK suicide vector. The constructed plasmids were named pJQ200SK-*chi60*-*Kmr* and pJQ200SK-*chi70*-*tetA*. The plasmids were transformed into *E. coli* S17-1 λ pir using the heat shock method. Recombinant *E. coli* was grown in LB medium with ampicillin (100 μg/mL) (Solarbio, Beijing, China) and kanamycin (50 μg/mL) (Solarbio, Beijing, China)/tetracycline (10 μg/mL) (Solarbio, Beijing, China) at 37 °C for 16 h, respectively. The positive clones were subjected to Beijing Genomics Institute for sequencing.

#### 5.4.3. Single Gene Knockout Mutants Screening

Recombinant *E. coli* S17-1 λ pir containing the suicide plasmid were grown in liquid LB medium at 37 °C. At the same time, *X. nematophila* HB310 was grown in LB medium at 28 °C. When liquid cultures were grown to an OD_600_ nm of 0.7, 1 mL cultures were harvested and washed three times using fresh LB medium, respectively. The cells were resuspended in 100 μL LB. Then, the *E. coli* S17-1 λ pir and *X. nematophila* were mixed in 1:3 ratio (20 μL *E. coli* S17-1 λ pir: 60 μL *X. nematophila*) and spotted onto LB agar plates. Plates were incubated for 3 h at 37 °C and then 28 °C overnight. The bacterial colonies were suspended in the liquid LB medium. The mixture was spread on the LB agar plates containing ampicillin and corresponding resistance. The single colony was transferred into liquid LB medium containing ampicillin and the corresponding resistance for culture. The culture was spread on the LB agar plates containing ampicillin and the corresponding resistance. The single colony was transferred into an LB medium (NaCl free) containing 6% sucrose (Solarbio, Beijing, China). When the transformants were obtained, the genomic DNA was isolated. The genotype of the mutant was confirmed by PCR and the products of PCR were subjected to the Beijing Genomics Institute for sequencing.

#### 5.4.4. Double Gene Knockout Mutant Screening

The Δ*chi70* was grown in LB medium (NaCl free) containing 6% sucrose with ampicillin and tetracycline at 28 °C, and the recombinant *E. coli* S17-1 λ pir containing *chi60*-*Kmr* gene was grown in liquid LB medium with kanamycin at 37 °C. The method of treatment and the culture conditions were consistent with the above method, except for the antibiotics in the medium from ampicillin and tetracycline to ampicillin, kanamycin and tetracycline.

### 5.5. Western Blot

The cells of *X. nematophila* HB310 were centrifuged (4 °C, 10,000 g, 10 min) from culture broth, washed three times with 10 mM PBS (pH 7.2), and suspended in PBS (adding 5 mL PBS per 200 mL bacterial solution). The bacterial cells were lysed by sonication (2 s on, 3 s off, 30 cycles) and centrifuged at 4 °C and 10,000 g for 30 min. The supernatant was collected and filtered with 0.22 μm membrane. The Tc toxins were isolated by precipitation with 85% saturated ammonium sulfate and concentrated using a Centriprep 100 ultrafiltration device with a molecular mass cutoff of 100 kDa (Millipore Corporation, Shanghai, China). The Tc toxins were separated by 6% native polyacrylamide gel electrophoresis (PAGE) using the LIUYI model DYCZ-24F dual vertical electrophoresis apparatus (LIUYI, Beijing, China). The fractions were monitored by a UV detector and collected using a fraction collector, then concentrated by a Centriprep 100 device. 

To verify the correctness of the gene knockout, the primary antibody (Chi60 and Chi70 chitinase antiserum) and commercial antibodies (Goat anti-mouse IgG, HRP conjugated) (CWBIO, Beijing, China) were used in the Western blot analysis, which was performed as previously described [46,47].

### 5.6. Assay for Insecticidal Activity of Tc Toxins

The insecticidal activity of Tc toxins from the WT, Δ*chi60*, Δ*chi70* and Δ*chi60*-*chi70* against the second instar larvae of *H. armigera* was determined. The Tc toxins of the WT, Δ*chi60* and Δ*chi70* were diluted to 2000, 1000, 500, 250, 125 and 62.5 μg/mL, while Tc toxins from Δ*chi60*-*chi70* were diluted to 32,000, 16,000, 8000, 4000, 2000, 1000 and 500 μg/mL. The method of treatment and the culture conditions were consistent with the above methods. The LC_50_ was calculated at 120 h after treatment.

### 5.7. Data Analysis

The significance of differences in the growth inhibition rate and corrected mortality of chitinase to *H. armigera* larvae were analyzed by independent samples *t*-test (SPSS v26.0 software). Mortality data were analyzed by Probit regression (SPSS v26.0 software) to calculate the LC_50_ for each protein and mixture, with corresponding confidence limits and slopes of regression lines.

## Figures and Tables

**Figure 1 toxins-14-00646-f001:**
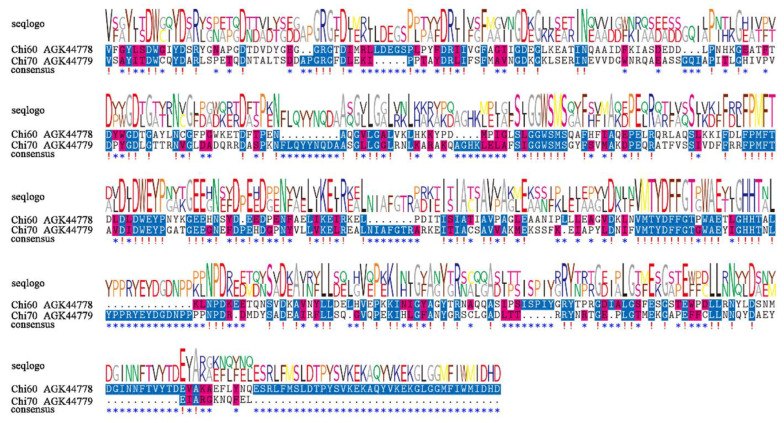
Sequence alignment of the conserved domains of Chi60 and Chi70 from *X. nematophila* HB310. !, conserved sites. *, differential sites.

**Figure 2 toxins-14-00646-f002:**
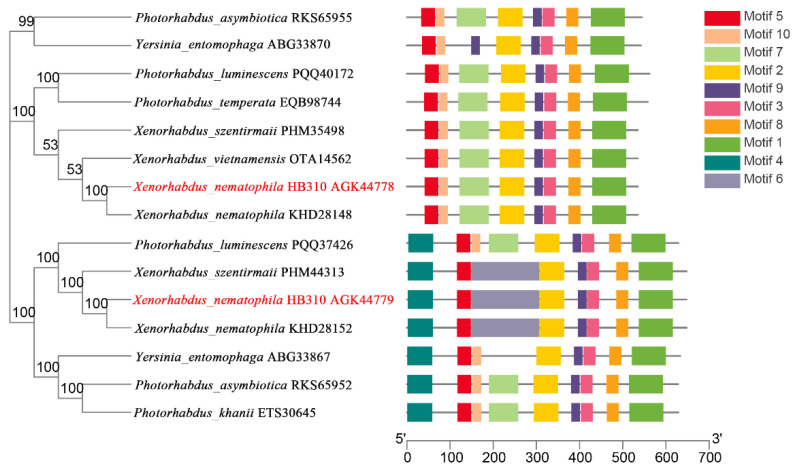
Phylogenetic relationships and composition of the conserved motif patterns. The phylogenetic tree was constructed based on the full-length sequences of chitinases using the MEGA 5.0 software. The sequence information for each motif was provided in Appendix A. The conserved motifs were displayed in different colored boxes, and the length of protein can be estimated using the scale at the bottom.

**Figure 3 toxins-14-00646-f003:**
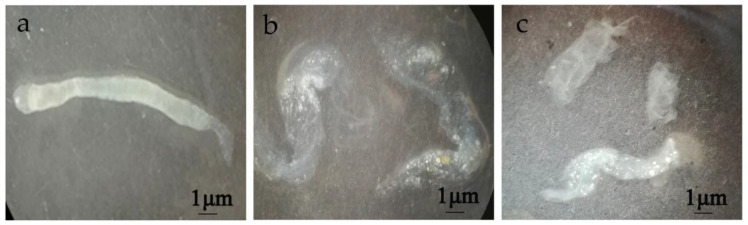
Changes of the peritrophic membrane of *H. armigera* after different treatments. (**a**) Treatment with PBS (pH 7.2) (negative control). (**b**) Treatment with Chi60—100 μg of Chi60 was fed per insect for 48 h, the peritrophic membrane broke into inelastic fragments. (**c**) Treatment with Chi70—100 μg of Chi70 was fed per insect for 48 h, the peritrophic membrane obviously broke into inelastic fragments.

**Figure 4 toxins-14-00646-f004:**
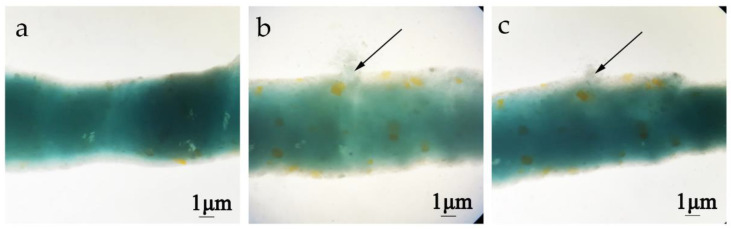
Changes in the peritrophic membrane permeability of *H. armigera* after different treatments. (**a**) Treatment with PBS (pH 7.2) and blue dextran 2000 (negative control). (**b**) Treatment with Chi60—100 μg of Chi60 and 10 μg blue dextran 2000 were fed per insect for 48 h, the peritrophic membrane displayed obvious exudation. (**c**) Treatment with Chi70—100 μg of Chi70 and 10 μg blue dextran 2000 were fed per insect for 48 h, the peritrophic membrane displayed obvious exudation.

**Figure 5 toxins-14-00646-f005:**
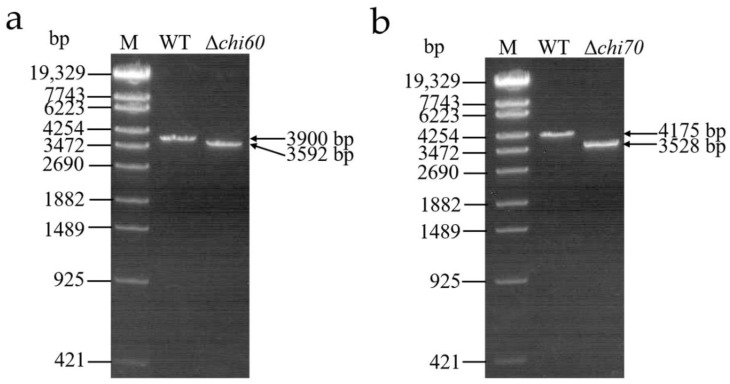
PCR identification of single gene knockout mutants. (**a**) PCR identification of *chi60* gene knockout mutants. (**b**) PCR identification of *chi70* gene knockout mutants. Lane M, λ-EcoT14 I digest DNA marker. Lane WT, the wild-type strain. Lane Δ*chi60*, the mutant strain with *chi60* gene knocked out. Lane Δ*chi70*, the mutant strain with *chi70* gene knocked out.

**Figure 6 toxins-14-00646-f006:**
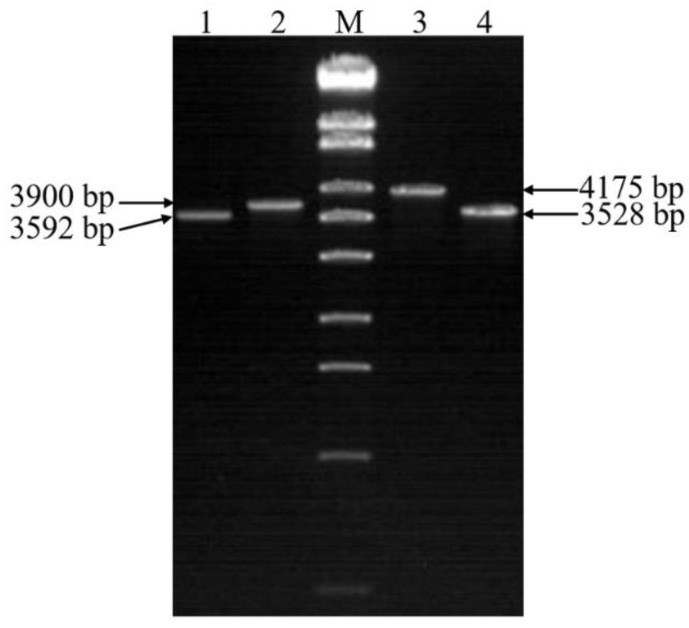
PCR identification of double gene knockout mutants. Lane M, λ-EcoT14 I digest (19,329, 7743, 6223, 4254, 3472, 2690, 1882, 1489, 925, 421, and 74 bp). Lane 1, the mutant strain with *chi60* gene knocked out. Lane 2 and lane 3, the wild-type strain. Lane 4, the mutant strain with *chi70* gene knocked out.

**Figure 7 toxins-14-00646-f007:**
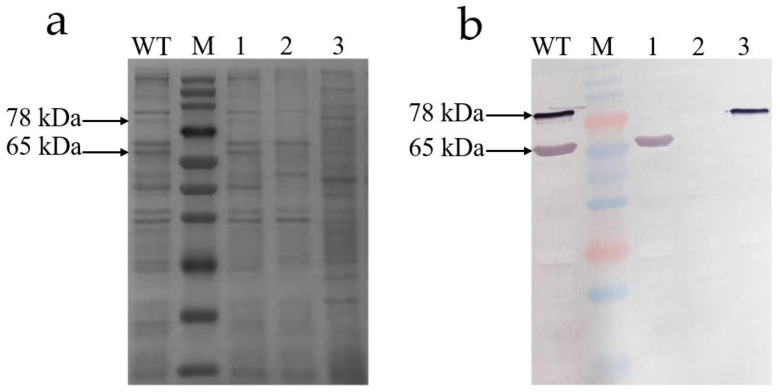
SDS-PAGE (12%) and Western blotting analysis of Tc toxins from WT, Δ*chi60*, Δ*chi70* and Δ*chi60*-*chi70*. (**a**) SDS-PAGE stained with Coomassie brilliant blue analysis. (**b**) Western blotting analysis. Lane M, multicolor prestained protein marker (250, 150, 100, 70, 50, 40, 35, 25, 20, 15, and 10 kDa). Lane WT, the wild-type strain. Lane 1, the mutant strain with *chi70* gene knocked out. Lane 2, the mutant strain with *chi60* and *chi70* gene knocked out simultaneously. Lane 3, the mutant strain with the *chi60* gene knocked out.

**Table 1 toxins-14-00646-t001:** The corrected mortality and growth inhibition rate of Chi60 and Chi70 against *H. armigera*.

Treatment	Growth Inhibition Rate (%)	Corrected Mortality (%)
Chi60	81.99 ± 2.42	4.17 ± 2.41
Chi70	90.51 ± 1.44 *	13.89 ± 1.39 *

*, Significant difference at *p* < 0.05 level by independent samples *t*-test.

**Table 2 toxins-14-00646-t002:** The synergistic effect of Chi60 and Chi70 with Tc toxins against *H. armigera*.

Treatment	LC_50_ (μg/mL)	95% CL	Slope ± SE	R^2^
Tc toxins	196.11	149.30–251.30	2.37 ± 0.33	0.99
Tc toxins + Chi60	185.85	151.75–224.32	3.00 ± 0.34	0.98
Tc toxins + Chi70	146.47	113.14–175.84	2.52 ± 0.31	0.99

LC_50_, lethal concentration to 50% of the population. 95% CL, 95% confidence limits. SE, standard error. R^2^, correlation coefficient.

**Table 3 toxins-14-00646-t003:** Insecticidal activity of Tc toxins from WT and mutant strains against *H. armigera*.

Treatment	LC_50_ (μg/mL)	95% CL	Slope ± SE	R^2^
WT	196.11	149.30–251.30	2.37 ± 0.33	0.99
Δ*chi60*	885.74	650.53–1342.22	1.68 ± 0.28	0.99
Δ*chi70*	757.25	551.29–1144.21	1.54 ± 0.22	0.99
Δ*chi60*-*chi70*	20,049.83	10,711.52–64,351.57	0.80 ± 0.16	0.99

LC_50_, lethal concentration to 50% of the population. 95% CL, 95% confidence limits. SE, standard error. R^2^, correlation coefficient. WT, wild-type strain. Δ*chi70*, the mutant strain with *chi70* gene knocked out. Δ*chi60*, the mutant strain with *chi60* gene knocked out. Δ*chi60*-*chi70*, the mutant strain with *chi60* and *chi70* genes knocked out simultaneously.

## Data Availability

Not applicable.

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
