# Peer review of "Insecticidal Activity of Chitinases from Xenorhabdus nematophila HB310 and Its Relationship with the Toxin Complex"

_toxins, 2022, doi:10.3390/toxins14090646_

Round 1
Reviewer 1 Report
In the manuscript, “Insecticidal Activity of Chitinases from Xenorhabdus nematophila and its Relationship with the Toxin Complex”, the authors try to demonstrate that the presence of chi60 and chi70 is indispensable for the toxicity of Tc toxins as an insecticide. There are some concerns about this manuscript.
Major comments:
1. Two bands of 78 kDa (Chi70) and 65 kDa (Chi60). However, 76 kDa and 65 kDa were labled in Fig. 5. And, in the Fig. 5b, the molecular weight of the band is higher than the expected 78 kDa (Chi70), how to explain it?
2. 166-169: “However, the inhibition of Chi60 was significantly lower than that of Chi70, which could be related to the inclusion of Chi60. After denaturation and renaturation, Chi60 could become a soluble protein, but the natural conformation could not be completely restored. Therefore, the inhibition of Chi60 in the larvae of H. armigera was lower than that of Chi70.” These statements appear in the discussion section, however, it should have citation references. In contrast, if the results described in the statements were established in the authors’ Lab, they should illustrate in the result section.
3. The legend should indicate the experimental conditions briefly to help readers more easily to understand.
Minor comments:
1. Fig. 3 “Lane M, λ-EcoT14 â… digest (TaKaRa, Beijing).” (TaKaRa, Beijing) should be deleted. The names of companies should be mentioned in Materials and Methods.
2. The size of markers should be labled in the Fig. 3, 4 and 5.
3. In Fig. 4, the arrowheads indicated the 5400 bp and 4175 bp. However, the migration distance in the 4175 bp was slower than the 5400 bp. Why? And this is not possible!
Author Response
Response to Reviewer 1 Comments
Dear Editors and Reviewers:
We wish to thank you for reviewing our paper. And, we also thank you for pointing out some problems to help us improve our work. Below, we present a point-by-point listing of all the comments that were addressed and how related changes were incorporated into the revised manuscript.
We would like to thank you once again, and hope this revised draft meets your expectations. We look forward to hearing more in due course.
Yours sincerely,
All co-authors
Point 1: Two bands of 78 kDa (Chi70) and 65 kDa (Chi60). However, 76 kDa and 65 kDa were labled in Fig. 5.
Response 1: We are very sorry for our incorrect writing and it is rectified in figure 7 (figure 5 in the original manuscript) of revised manuscript. The size of the recombinant Chi70 protein is 78 kDa, which had been reported in the reference 9. (Line 208, P7)
[9] Liu, J.; Nangong, Z.; Zhang, J.; Song, P.; Tang, Y.; Gao, Y.; Wang, Q. Expression and characterization of two chitinases with synergistic effect and antifungal activity from Xenorhabdus nematophila. World J. Microbiol. Biotechnol. 2019, 35, 106.
Point 2: And, in the Fig. 5b, the molecular weight of the band is higher than the expected 78 kDa (Chi70), how to explain it?
Response 2: Revised. This may be caused by the crooked placement position of PVDF membrane during the membrane transfer process. The voltage is high and the speed is fast when the film is transferred by iBlot 2 PVDF Mini Stacks. figure 5b shows that the right band is higher than the left. By consulting the previous pictures, a new figure was selected and replaced. (Line 208, P7)
Point 3: 166-169: “However, the inhibition of Chi60 was significantly lower than that of Chi70, which could be related to the inclusion of Chi60. After denaturation and renaturation, Chi60 could become a soluble protein, but the natural conformation could not be completely restored. Therefore, the inhibition of Chi60 in the larvae of H. armigera was lower than that of Chi70.” These statements appear in the discussion section, however, it should have citation references. In contrast, if the results described in the statements were established in the authors’ Lab, they should illustrate in the result section.
Response 3: Revised. Chi60 was an inclusion bodies protein, which the process of denaturation and renaturation had been reported in reference 9. It had been added to the corresponding position in the revised manuscript. (Line 245-247, P8)
[9] Liu, J.; Nangong, Z.; Zhang, J.; Song, P.; Tang, Y.; Gao, Y.; Wang, Q. Expression and characterization of two chitinases with synergistic effect and antifungal activity from Xenorhabdus nematophila. World J. Microbiol. Biotechnol. 2019, 35, 106.
Point 4: The legend should indicate the experimental conditions briefly to help readers more easily to understand.
Response 4: Thanks for your constructive comments, which is of great help to improve the quality of our manuscript. It has been modified in the revised manuscript. (Line 88-89, P3. Line 101-105, P3. Line 126-127, P4. Line 138-142, P5. Line 149-153, P5. Line 183-187, P6. Line 194-197, P6-7. Line 209-214, P7. Line 226-229, P8.)
Point 5: Fig. 3“Lane M, λ-EcoT14 â… digest (TaKaRa, Beijing).” (TaKaRa, Beijing) should be deleted. The names of companies should be mentioned in Materials and Methods.
Response 5: Thanks for your advice. We have removed the names of companies in the all figures. (Line 185, P6. Line 195, P6. Line 212, P7.)
Point 6: The size of markers should be labled in the Fig. 3, 4 and 5.
Response 6: Thanks for your comments. The size of markers had been added in figure 5, 6, and 7 (figure 3, 4 and 5 in the original manuscript) of revised manuscript. There were bands on both sides of the marker in figure 6 and 7, so the molecular weights of the molecular marker were added to the legend. (Line 195, P6. Line 211-212, P7.)
Point 7: In Fig. 4, the arrowheads indicated the 5400 bp and 4175 bp. However, the migration distance in the 4175 bp was slower than the 5400 bp. Why? And this is not possible!
Response 7: Thank you for your reminding. The 5400bp in figure 4 should be corrected into 3900bp. It was mistaken due to lack of consistency check. Luckily, the descriptions in manuscript that the fusion fragment size of chi60-Kmr is 3900 bp were right. (Line 193, P6.)

Reviewer 2 Report
The paper Insecticidal activity of chitinases from Xenorhabdus nematophila and its relationship with the toxin complex describe scientific research related to the mode of action of Chitinase
The topic fit the journal
The topic is quite studied in the specific area but the investigations are needed to clarify mode of action and to develop new biopesticides.
The introduction is presenting sufficiently the literature state of the art. I suggest to better explain and to underline the novelty of the present paper compared to the previously published papers. I suggest to check the most novel papers published on similar topic and to clearly explain to reader what are the novelty you want to present.
The materials and methods is sufficiently clear.
Results are describing quite a large amount of experimental data, and authors applied many state of the art approches of biothecnology in appropiate mode. I suggest to improve the result discussion and to add small summary of the expected result at the beginning of each small chapter reated to each experimental part. As example for the 2.4 small sentence explaining why you are doing Δchi60- and Δchi70- Screening so helping reder to follows your logic of the experiments
To make more incisive your discussion I suggest to start with small summary of the obtained results underlying what are the novelties and then proceed as you did with discussion of the bibliographyc references.
I think that the paper is interesting and present large amount of experimental work
Authors can implement trying to make more visible their novel results and the significance in the area of the development of new biopesticides.
Some minor comments are also included in the pdf

Author Response
Response to Reviewer 2 Comments
Dear Editors and Reviewers:
We wish to thank you for reviewing our paper. And, we also thank you for pointing out some problems to help us improve our work. Below, we present a point-by-point listing of all the comments that were addressed and how related changes were incorporated into the revised manuscript.
We would like to thank you once again, and hope this revised draft meets your expectations. We look forward to hearing more in due course.
Yours sincerely,
All co-authors
Point 1: The topic is quite studied in the specific area but the investigations are needed to clarify mode of action and to develop new biopesticides.
Response 1: Thanks for your advice. We will investigate further and explore the mode of action. At the same time, we will also pay more attention to the development of new biological pesticides, and strive to develop a good biological pesticide. (Line 292-298, P9)
Point 2: The introduction is presenting sufficiently the literature state of the art. I suggest to better explain and to underline the novelty of the present paper compared to the previously published papers. I suggest to check the most novel papers published on similar topic and to clearly explain to reader what are the novelty you want to present.
Response 2: Such a suggestion is indeed valuable. The previously published articles analyzed the structure and mechanism of Tc toxins and the function of Tc toxins single protein. One of these articles described the structure of Tc toxins from Yersinia entomophaga in relation to chitinases. It reported that the structure of Tc toxins could not be formed after knocking out two chitinase genes simultaneously.
In my research, a single chitinase gene was knocked out first, and then two chitinase genes were simultaneously knocked out to verify the effect of chitinases on the insecticidal activity of Tc toxins. It can not only lay the foundation for the development and utilization of insecticides of entomopathogenic nematode symbiotic bacteria, but also provide candidate genes for transgenic insect-resistant plants. (Line 79-81, P2)
[1] Landsberg, M.J.; Jones, S.A.; Rothnagel, R.; Busby, J.N.; Marshall, S.D.; Simpson, R.M.; Lott, J.S.; Hankamer, B.; Hurst, M.R. 3D structure of the Yersinia entomophaga toxin complex and implications for insecticidal activity. Proc. Natl. Acad. Sci. USA 2011, 108, 20544-20549.
[2] Busby, J.N.; Landsberg, M.J.; Simpson, R.M.; Jones, S.A.; Hankamer, B.; Hurst, M.R.; Lott, J.S. Structural analysis of Chi1 chitinase from Yen-Tc: The multisubunit insecticidal ABC toxin complex of Yersinia entomophaga. J. Mol. Biol. 2012, 415, 359-371.
[3] Lang, A.E.; Konukiewitz, J.; Aktories, K.; Benz, R.; TcdA1 of Photorhabdus luminescens: Electrophysiological analysis of pore formation and effector binding. Biophys. J. 2013, 105, 376-384.
[4] Gatsogiannis, C.; Lang, A.E.; Meusch, D.; Pfaumann, V.; Hofnagel, O.; Benz, R.; Aktories, K.; Raunser, S. A syringe-like injection mechanism in Photorhabdus luminescens toxins. Nature 2013, 495: 520-523.
[5] Gatsogiannis, C.; Merino, F.; Prumbaum, D.; Roderer, D.; Leidreiter, F.; Meusch, D.; Raunser, S. 2016. Membrane insertion of a Tc toxin in near-atomic detail. Nat. Struct. Mol. Biol. 2016, 23, 884-890.
[6] Gatsogiannis, C.; Merino, F.; Roderer, D.; Balchin, D.; Schubert, E.; Kuhlee, A.; Hayer-Hartl, M.; Raunser, S. Tc toxin activation requires unfolding and refolding of a beta-propeller. Nature 2018, 563, 209-213.
[7] Meusch, D.; Gatsogiannis, C.; Efremov, R.G.; Lang, A.E.; Hofnagel, O.; Vetter, I.R.; Aktories, K.; Raunser, S. Mechanism of Tc toxin action revealed in molecular detail. Nature 2014, 508, 61-65.
[8] Roderer, D.; Schubert, E.; Sitsel, O.; Raunser, S. Towards the application of Tc toxins as a universal protein translocation system. Nat. Commun. 2019, 10, 5263.
[9] Sheets, J.J.; Hey, T.D.; Fencil, K.J.; Burton, S.L.; Ni, W.; Lang, A.E.; Benz, R.; Aktories, K. Insecticidal toxin complex proteins from Xenorhabdus nematophilus: Structure and pore formation. J. Biol. Chem. 2011, 286, 22742-22749.
[10] Roderer, D.; Raunser, S. Tc toxin complexes: Assembly, membrane permeation, and protein translocation. Annu. Rev. Microbiol. 2019, 73, 247-265.
Point 3: Results are describing quite a large amount of experimental data, and authors applied many state of the art approaches of biotechnology in appropriate mode. I suggest to improve the result discussion and to add small summary of the expected result at the beginning of each small chapter related to each experimental part. As example for the 2.4 small sentence explaining why you are doing Δchi60- and Δchi70- Screening so helping reader to follows your logic of the experiments
Response 3: Thank you for your suggestion. we have added a small summary of the expected result at the beginning of each small chapter related to each experimental parts in the revised manuscript. (Line 107-109, P4. Line 117-119, P4. Line 129-131, P4. Line 143-144, P5. Line 199-200, P7. Line 216-218, P7.)
Point 4: To make more incisive your discussion I suggest to start with small summary of the obtained results underlying what are the novelties and then proceed as you did with discussion of the bibliographyc references.
Response 4: Good point. After careful consideration, we think that your suggestion can make the discussion even more incisive. In this paragraph, the effect of chitinase gene knockout on the structure of Tc toxins has been discussed, and a supplementary explanation for the research that may continue in the futural also has been illustrated.
See below: “The main research at present is the effect of chitinase gene knockout on the insecticidal activity of Tc toxins. The next step is to further study the effect of chitinase gene knockout on the structure of Tc toxins. In addition, it is also necessary to pay attention to a development of biological pesticide, and effectively develop the Tc toxins into the new biological pesticides and produce it on a large scale. At the same time, further exploration of transgenic insect-resistant plants with Tc toxin genes is needed.” (Line 292-298, P9)
Point 5: I think that the paper is interesting and present large amount of experimental work. Authors can implement trying to make more visible their novel results and the significance in the area of the development of new biopesticides.
Response 5: Thanks for your suggestion. We next plan to study the effect of chitinase gene knockout on the structure of Tc toxins, so as to further clarify the effect of chitinase gene knockout on the structure and function of Tc toxins. At the same time, we are preparing to clone the full-length sequence of Tc toxins, and try to transfer it into plants through transformation mediation. (Line 292-298, P9)

Reviewer 3 Report
Dear Authors, the major problem of this ms is the results especially Insecticidal Activity of Chitinases and Synergistic Effect with Tc Toxin. I believe this part is very bad because you dont present solid results.
This ms is this section doesnot have results. I feel you must do more experiments.

Author Response
Response to Reviewer 3 Comments
Dear Editors and Reviewers:
We wish to thank you for reviewing our paper. And, we also thank you for pointing out some problems to help us improve our work. Below, we present a point-by-point listing of all the comments that were addressed and how related changes were incorporated into the revised manuscript.
We would like to thank you once again, and hope this revised draft meets your expectations. We look forward to hearing more in due course.
Yours sincerely,
All co-authors
Point 1: This ms is this section doesnot have results. I feel you must do more experiments.
Response 1: Thank you. In previous studies, chitinases or toxin proteins were generally tested for their insecticidal activities. If there were two or more proteins, the synergistic effect would be determined. In this study, chitinase alone could significantly inhibit the growth of Helicoverpa armigera, but the lethal effect on H. armigera was very low. In our previous research, we tried to add different amounts (100μg, 300μg, 500μg, 1000μg, and 1500μg) of chitinase to Tc toxins to determine its insecticidal activity, and finally selected the most suitable amount (1000μg) of chitinase in this study. In addition, we also determined the synergistic effect of chitinase against Bt HD73 strain and Bt Cry1Ac toxins. Similar to the result of this study, Chi70 has an obviously synergistic effect with Bt Cry1Ac and Bt HD73 strain. At present, due to the impact of the COVID-19, some work has been temporarily put on hold. In the future, we will continue to do more experiments that about the effects of other concentrations of chitinase on the insecticidal activity of Tc toxins, and strive to screen out the combinations with better activity. (Line 106-127, P4)
[1] Wiwat, C.; Thaithanun, S.; Pantuwatana, S.; Bhumiratana, A. Toxicity of chitinase-producing Bacillus thuringiensis ssp. kurstaki HD-1 (G) toward Plutella xylostella. Journal of Invertebrate Pathology. 2000, 76, 270-277.
[2] Lee, M.K.; Curtiss, A.; Alcantara, E.; Dean, D.H. Synergistic effect of the Bacillus thuringiensis toxins CryIAa and CryIAc on the gypsy moth, Lymantria dispar. Applied and Environmental Microbiology. 1996, 62:583-586.
[3] Yang, J.; Quan, Y.; Sivaprasath, P.; Shabbir, M.Z.; Wang, Z.; Ferré, J.; He, K. Insecticidal activity and synergistic combinations of ten different Bt toxins against Mythimna separata (Walker). Toxins (Basel). 2018, 10, 454.
[4] Nangong, Z.; Wang, Q.Y.; Song, P.; Hao, J.; Yang, Q.; Wang L.Y. Synergism between Bacillus thuringiensis and Xenorhabdus nematophila against resistant and susceptible Plutella xylostella (Lepidoptera: Plutellidae). Biocontrol Science & Technology. 2016, 26, 1-22.
Point 2: In table 1, why corrected? control mortality?
Response 2: Thank you. Corrected mortality refers to the mortality of the drug treatment group corrected by the natural mortality of the blank control group. In the process of efficacy trials, the tested biological populations often suffer from natural death due to natural enemy predation, parasitism or weak individual viability. Therefore, the PBS as a blank control group to correct the mortality rate of chitinase treatment group in this research. (Line 114-116, P4)
In addition, in terms of data analysis, we consulted relevant statistical experts, and also checked and referred to the analysis methods involving corrected mortality, we rechecked the data.
[1] Shi, X.; Qiao, K.; Li, B.; Zhang, Sh. Integrated management of Meloidogyne incognita and Fusarium oxysporum in cucumber by combined application of abamectin and fludioxonil. Crop Protection. 2019, 126, 104922.
[2] Jiang, Q.; Xie, Y.; Peng, M.; Wang, Z.; Li, T.; Yin, M.; Shen, J.; Yan, S. A nanocarrier pesticide delivery system with promising benefits in the case of dinotefuran: strikingly enhanced bioactivity and reduced pesticide residue. Environmental Science: Nano. 2022, 9, 988-999.
[3] Zhou, X.; Liang, W.; Zhang, Y.; Ren, Z.; Xie, Y. Effect of earthworm Eisenia fetida epidermal mucus on the vitality and pathogenicity of Beauveria bassiana. Scientific Reports. 2021, 11, 13915.
Point 3: In table 2, why this synergistic?
Response 3: Thank you. “Synergistic effects are the combined effects of at least two medicines that have a bigger impact than any of them could have on its own”. Here, we want to reflect the effect of recombinant chitinase on the insecticidal activity of Tc toxins by the insecticidal activity of the mixture of Tc toxins and chitinase on Helicoverpa armigera. (Line 125-127, P4)
In this regard, we also refer to other similar researches in related fields:
[1] Nangong, Z.; Wang, Q.Y.; Song, P.; Hao, J.; Yang, Q.; Wang L.Y. Synergism between Bacillus thuringiensis and Xenorhabdus nematophila against resistant and susceptible Plutella xylostella (Lepidoptera: Plutellidae). Biocontrol Science & Technology. 2016, 26, 1-22.
[2] Broderick, N.A.; Goodman, R.M.; Raffa, K.F.; Handelsman, J. Synergy Between Zwittermicin A and Bacillus thuringiensis subsp. kurstaki Against Gypsy Moth (Lepidoptera: Lymantriidae). Environmental Entomology. 2000, 29, 101-107.
[3] Peng, D.; Chai, L.; Wang, F.; Zhang, F.; Ruan, L.; Sun, M. Synergistic activity between Bacillus thuringiensis Cry6Aa and Cry55Aa toxins against Meloidogyne incognita. Microbial Biotechnology. 2011, 4, 794-798.
Point 4: In line 113, why chi60 and chi70 italics?
Response 4: Thank you. The chi60 and chi70 represent the names of two chitinase genes from X. nematophila HB310 in the manuscript. Therefore, they expressed in italics. (Line 158, P5)
Point 5: In discussion section, full name classification of species should be added.
Response 5: Thank you very much for your professional suggestions. The full name classification of species had been added in the revised manuscript. (Line 72, P2. Line 235-266, P7-8)

Reviewer 4 Report
The manuscript reports the study of two chitinases from Xenorhabdus nematophila for verify their insecticidal activity. These chitinase are located on the gene cluster encoding Tc toxins.
In my opinion, the manuscript has enough novelty for its publication. At the same time, some points need to be implemented.
Main points:
1. In figure 2, the authors must insert the molecular weights of the molecular marker;
2. This reviewer would be pleased to see in questions response the raw western blot membrane of Figure 5B;
3. The authors must add some structural information of the two chitinases (identity, similarity or other) considering the different effect of the two chitinases;
Finally, in Materials and methods not all indications on the suppliers of the materials are indicated.
Author Response
Response to Reviewer 4 Comments
Dear Editors and Reviewers:
We wish to thank you for reviewing our paper. And, we also thank you for pointing out some problems to help us improve our work. Below, we present a point-by-point listing of all the comments that were addressed and how related changes were incorporated into the revised manuscript.
We would like to thank you once again, and hope this revised draft meets your expectations. We look forward to hearing more in due course.
Yours sincerely,
All co-authors
Point 1: In figure 2, the authors must insert the molecular weights of the molecular marker;
Response 1: Thank you very much for your suggestions. There is no molecular marker in figure 4 (Figure 2 in the original manuscript) of revised manuscript. The molecular weights of the molecular marker had been added in figure 5, 6, and 7 (figure 3, 4 and 5 in the original manuscript) of revised manuscript. (Line 195, P6. Line 211-212, P7.)
Point 2: This reviewer would be pleased to see in questions response the raw western blot membrane of Figure 5B
Response 2: Thank you very much for your recognition. Figure 5B in the original manuscript may be caused by the crooked placement position of PVDF membrane during the membrane transfer process. The voltage is high and the speed is fast when the film is transferred by iBlot 2 PVDF Mini Stacks. Figure 5b shows that the right band is higher than the left. By consulting the previous pictures, a new figure was selected and replaced. (Line 208, P7)
Point 3: The authors must add some structural information of the two chitinases (identity, similarity or other) considering the different effect of the two chitinases;
Response 3: Thanks for your constructive comments, which is of great help to improve the quality of our manuscript. The sequence alignment of Chi60 and Chi70, phylogenetic relationship between chitinases from Xenorhabdus nematophila HB310 and other bacteria, and architecture of the conserved motif patterns were added in the revised manuscript. (Line 83-105, P2-3. Line 328-341, P9-10)
Point 4: Finally, in Materials and methods not all indications on the suppliers of the materials are indicated.
Response 4: Revised. The suppliers of the materials in Materials and methods had been added in the revised manuscript. (Line 369-446, P10-12)

Round 2
Reviewer 2 Report
All raised comments were satisfactorily solved
i think the paper is implemented and can be accepted
Reviewer 3 Report
Accept in present form
Reviewer 4 Report
In my opinion after the corrections made the revised manuscript can be accepted.